
**Review article: Brief history of volcanic risk in Neapolitan area (Campania,**
**Southern Italy): a critical review**
**Stefano Carlino**
Istituto Nazionale di Geofisica e Vulcanologia, Sezione di Napoli, Osservatorio Vesuviano
**Corrispondence**: stefano.carlino@ingv.it
**Abstract**
The presence of three active volcanoes (Vesuvius, Campi Flegrei and Ischia Island) along the coast
of Naples did not constrained the huge expansion of the urbanized zones around them. On the
contrary, since Greek-Roman era, volcanoes have been an attractor for people who colonized
Campania region. Stable settlements around Vesuvius, Campi Flegrei caldera and the Island of Ischia
were progressively enlarged, reaching the maximum growth-rate between 1950 and 1980. Between
1982 and 1984, Neapolitan people faced the last and most dramatic volcanic crises, occurred at Campi
Flegrei (Pozzuoli), without an eruption. Since that time, volcanologists have focused the attention on
the problem of risk associated to eruptions in Neapolitan area, but a systematic strategy to reduce the
very high volcanic risk of this area still lacks. A brief history of volcanic risk in Neapolitan district is
here reported, trying to obtain new food for thought for the scientific community which works to the
mitigation of volcanic risk of this area.
**Keywords**: Neapolitan volcanoes, volcanic risk, volcanic hazard, risk mitigation, human settlements.
## 1.  Introduction
The district around Naples is one of the most-risky volcanic area in the World, due to the presence of
three active volcanoes, the Vesuvius, the Campi Flegrei caldera and the Island of Ischia, which are
inhabited by more than 1,500,000 people, directly exposed to the risk (Alberico et al., 2011; Carlino,
2019) (Fig. 1). These volcanoes have been capable to generate a wide range of eruptions, from gently
lava flow to catastrophic events and were active in historical times (the last eruption occurred in 1944
at Vesuvius, in 1538 at Campi Flegrei and in 1302 at Ischia Island). Larger eruptions at Vesuvius
generated the devastation of entire sectors of territory around the volcano, up to a distance of 10 km
to 20 km from the vent, such as in the case of 79 AD (Pompei) and 3,800 (Avellino) events,
respectively. At least two large caldera-forming eruptions occurred at Campi Flegrei (the Campania
Ignimbrite CI, ~39 ka, and the Neapolitan Yellow Tuff NYT, ~15 ka) which involved the whole
Campania Plain, such as the case of the CI event. At Ischia, a large eruption occurred about 55 ka



ago, while the subsequent activity was mostly confined inside the island (Piochi et al., 2005; De Vivo
et al., 2006; Mastrolorenzo et al., 2006; de Vita et al., 2010). In figure 2 a sketch of the eruptive
history of Vesuvius, Campi Flegrei and Ischia is reported (from Piochi et al., 2005).
In accounting the present risk situation of Neapolitan volcanoes, an historical and scientific approach
is necessary in order to understand the reasons why so many people are nowadays living in a such a
hazardous area, which was repeatedly hit by eruptions. On one hand, volcanoes themselves
represented the main attractor for people living in Neapolitan district, because their activity produced
fertile soils for farming, hot waters for human recreation, raw materials and natural inlets along the
coast for sea navigators (Carlino et al., 2010a; Scarpati et al., 2016). On the other hand, volcanic
activity generated large devastations of the area and many victims (Scarpati et al., 2013). The city of
Naples itself stands on various volcanic centers and, in particular, on the extended deposits of the
NYT eruption (~15 ka) which generated the collapse of the present Campi Flegrei caldera (Isaia et
al., 2009; Scarpati et al., 2013), the eastern rim of which is the site where the most beautiful and
elegant neighbored of the city (the *Posillipo hill*) stands (Fig. 3).
During the long history of relations between humans and Neapolitan volcanoes, few important
milestone events must be mentioned: Pompei 79 AD eruption, reconstructed by the letters of Plinian
the Younger; the eruption of Vesuvius of 1631which, after almost 500 years of quiescence, opened a
long period of continuous volcanic activity which ended in 1944; the systematic exploration of
Pompei (buried by the 79 AD event) starting from 1748; the foundation of "Osservatorio Vesuviano"
(Vesuvius Observatory), under the Bourbons domination, in 1841; the eruption of Vesuvius in 1944
which closed the activity of the volcano; the unrests crisis occurred at Campi Flegrei caldera in 1970-
72 and 1982-84 (Barberi et al., 1984; Giacomelli et al., 2003; Scandone et al., 2008; Perrotta and
Scarpati, 2009; Cubellis et al., 2015). In particular, in this paper the latter two crises at Campi Flegrei
will be discussed, since they occurred during an important moment of challenges in the field of the
Earth Science and during the period of the improvement of volcanoes monitoring networks and of
the policies for management and prevention of the risks in Neapolitan area (Carlino, 2019). Starting
form that time, the problem of volcanic hazard and risk in Neapolitan area has been systematically
treated by many authors, trying to quantify the equation of the risk: *risk = hazard x vulnerability x*
*exposed value* (see Blong, 1996 and references therein). A larger part of the studies has been aimed
to assess the hazard and, to a lesser extent, the risk (see for instance Scandone et al., 1993; Petrosino
et al., 2004; Mastrolorenzo et al., 2006) and the risk perception of communities exposed to potential
volcanic activity (Carlino et al., 2008; Ricci et al., 2013). Other authors have debated the criteria used
to identify the most risked area in Neapolitan volcanic district (e.g. the red zones) criticising the
emergency plan of Vesuvius, or proposing an alternative perspective to reduce the risk (Dobran, 2000,



2007; Matsrolorenzo et al., 2006; De Vivo et al., 2010; Rolandi, 2010; De Natale et al., 2020).
Although the very high risk reached in this district, only in recent time (starting from the early 2000)
a number of attempts to reduce its exposed values has been carried on, but unsuccessfully. Possibly,
a more general analysis, from both historical and scientific point of view, to understand the reasons
why the attempts to reduce the volcanic risk in Neapolitan area have systematically failed is
necessary. It is not intention of this paper to face such a complex issue, which deserves a wider, longer
and multidisciplinary discussion, but a thought about this topic is required. In this paper, it is reported
a brief history of volcanic risk in Neapolitan area, and an account of recent studies and policies
adopted to reduce the risk. As it will be showed, new proposals to mitigate the volcanic risk of this
area could be ineffective if we do not take into account the reasons why previous tentatives to reduce
the volcanic risk in Neapolitan area have failed. Furthermore, it is important to define, as more clearly
as possible, the role of volcanologists in facing volcanic emergency and risk education policies in this
high urbanized area.

## 2. The progressive human settlement of Neapolitan volcanoes
The history of risk of Neapolitan volcanoes begins before the birth of Christ, when the first stable
population settled the plain along Vesuvius and the Campi Flegrei caldera (Pappalardo, 2007). The
great Greek geographer Strabo (64 B.C.-19 A.D) reported in his work "Geography" one of the first
description of the Campania Plain and surroundings and denoted the splendor of these places,
dominated by the presence of Vesuvius and bordered by mountains which extended along the sea
forming the Gulf of Naples (Strabone, XIV-XXIII A.D.). Otherwise, it would seem that the first and
most ancient human settlements in Campania date back to the Palaeolithic, mainly along the coasts
of the Sorrento Peninsula. As far as we know, a first evidence of the disruption of human activity due
to volcanic eruption in this area dates back about 3,800 years (Mastrolorenzo et al., 2006). This is in
fact the age of an ancient Bronze Age village near Nola, about 11 km north of Mount Vesuvius, where
archaeological excavations uncovered a human village with many findings in a state of excellent
conservation. It was a massive explosive eruption of Vesuvius (the Avellino eruption, 3,800 years
ago) that sealed the village beneath hot ash (Mastrolorenzo et al., 2006), in a fate similar to what
happened in Pompeii a few thousand years later. That was the time when the natural environment of
Vesuvius showed a less friendly face, and humankind was confronted with unexpected adversities.
In fact, the geology and the landscape of Campania were the main attractions for the populations that
colonized this area, that Romans later called "Campania felix" (from Latin felix = lucky, happy)
(Montone, 2010). The expression derives not only from the beauty of the places, but also from the
fertility of the soil coming from the volcanic activity, the presence of streams and the gentleness of





climate. The broad river and coastal plains, the modest mountain ranges overlooking them, the steam
and the various volcanic areas, the thermal waters and natural coastal inlets to protect sailors, all
combined together to transform the area into the crossroads of different civilizations (Carlino, 2019).
The Campi Flegrei area is also linked to the myth, possible due to the suggestion recalled by the
continuous emission of hot steam and the boiling of mud pots. It was there, along the Lake of Averno
(a volcanic crater close to the city of Pozzuoli) that the ancients placed the cave of the Cumaean Sibyl
(motioned in the famous literary work "L'Eneide" of Virgilio) and the entrance to the afterlife
(Azcuy, 2013). This crater lake exhaled vapors and volcanic gases that probably kept some animals
away, from which it got its Greek name, aoèrnov, that is, "without birds". Following the migration
of Etruscan population, from central Italy to Campania plain, from the 9th to the 5th century B.C.,
the first early urban centers were established (Maiuri, 1957). These populations predominantly settled
in the fertile lowlands of the Campanian Plain, along the rivers or close to the river-mouths. With the
arrival of Greeks, and the development of the maritime trade, the inhabitants of Campania migrated
towards coastal areas and began to settle in the volcanic areas of Ischia (called "Pithecusae") and later
of Campi Flegrei and Vesuvius (D'Ascia, 1867). Greeks arrived between the 9th and the 8th centuries
B.C., from a long and narrow island close to the coast of modern-day south-east Greece, namely
Euboea. On the Phlegrean side, ancient signs of stable habitation dating to a period between the 7 and
6th centuries B.C., were found in the Rione Terra, the old town of the present-day Pozzuoli
(Pappalardo, 2007). The historical center of this town stands on a small volcanic promontory that at
that time played host to a modest Cumaean mooring. Between 529 and 528 B.C. some Samnite exiles,
banned by the tyrant Polycrates, founded a colony on the promontory with the rigid name of
Dikaiarchia, meaning 'Just Government', integrated into a territory still controlled from Cumae
(Annecchino, 1996). In 194 B.C. the Romans transformed this small colony into a town called
Puteolis (hereafter Pozzuoli), thus named for its abundance of thermal springs. The town soon became
an imposing port and warehousing area for large quantities of foodstuffs. Before that, Greeks moved
eastwards, establishing the first inhabited elements of the city of Naples (called Pharthenophe),
between Mount Echia, an upland of volcanic origin, and the island of Megaride where Castel dell'Ovo
stands today (Ghirelli, 2015). The Greek population faced with the hazard of volcanoes in the island
of Ischia. In fact, their migration from Ischia towards the coast of Campania was possibly influenced
by the eruptions in the western and southern parts of the island that followed from the fifth century
B.C. onwards. Amidst the lavas and the ash of the fifth century B.C. eruption and close to the port of
Ischia, an old ground level was excavated containing potsherds and other archaeological finds from
the 6 and 5th centuries B.C., demonstrating the existence of an ancient Greek settlement destroyed in
the eruption (Carlino et al., 2010a). It was Strabo to bear witness to the eruptions in the Greco-Roman
era, writing: "......*in ancient times a series of extraordinary events took place on the island of*
*Pithecusae. [...] when Mount Epomeo, which rises in the middle of the island, was shaken by*


*earthquakes and erupted fire and (again) swept away everything that lay between itself and the shore*
*and into the sea. At the same time a part of the ground, reduced to ash and thrown upwards, fell back*
*onto the island like a maelstrom and the sea retreated for a distance of three stadia (about 500 m)*
*and, flowing back shortly afterward, flooded the island, extinguishing the fire. Such was the deafening*
*noise that the inhabitants of the mainland fled from the coast to the inner regions of Campania.*" The
towns of Naples and Pozzuoli, and the villages of the Vesuvius area, such as Pompeii, were expanding
rapidly, knowing about the disasters of the Roman era, but rapidly having to deal with the adverse
forces generated by the volcanic nature of the area. While in historical times (starting from the former
civilized human settlements), the Campi Flegrei caldera and the island of Ischia generated small
eruptions, the Vesuvius, on the contrary, demonstrated its power with the 79 A.D. eruption which
seriously affected the cities of Pompei and Ercolano and the southern part of the volcano (Giacomelli
et al., 2003). During the longest period of expansion of the Western Roman Empire, the cities around
the volcanoes had expanded progressively. The volcanic activity of Ischia of the early centuries
before Christ and its insular nature had, however contained its demographic expansion. On the other
side, the quiescence of the Campi Flegrei in eruptive terms did not mean that the volcanic nature of
these places had been forgotten, the continuous puffs of steam and the hot thermal springs being a
clear sign of that. But, in the minds of people at least, the hostile nature of these places, sometimes
sinister, was associated with the mood of gods, and not the actual nature of the area itself (Carlino,
2019). In this emerges the vision of the natural disaster as a divine punishment for humankind, a
vision which remained rooted in the culture of people up to the 17[th] century. Starting from Galileo
Galilei (1564-1642) era, a gradual change of the approach to the study of the Earth Science and the
risk related to natural phenomena took place.
A crucial moment in the history of volcanic risk in Neapolitan area took place in 1631 when, after a
long period of quiescence, Vesuvius awoke with an explosive (sub-plinian) eruption, beginning an
almost continuous eruptive activity that only ceased in 1944 at the end of World War II (Rosi et al.,
1993; Kilburn and McGuire, 2001). However, here too a theological meaning was attributed to this
calamitous event, as an expiation of punishments and, in this sense, the eruption of 1631 represented
a symbolic event, which affected, in the coming centuries not only volcanology but also other
political, sociological, literary, and above all, religious disciplines (Scarth, 2009). Although the 17[th]
century was still dominated by Aristotelian culture, it was also the beginning of its end as a result of
the works of the Galileans and Cartesians (Fiorentino, 2015). It was a period of great cultural
transformations, with new impulses in the field of scientific research coming from the introduction
of the experimental method by Galileo Galilei (Rossi, 2020). A further support and impetus to the
scientific revolution was provided by the foundation of the Royal Society of London in 1662 and of
Acadèmie Royale des Sciences in Paris.





Actually, the eruption of 1631 of Vesuvius was the first event which focused the attention on the
problem of volcanic risk. In fact, the suggestion to mitigate the volcanic risk at Vesuvius was formally
proposed for the first time by the viceroy of Naples, Emmanuele Fonseca, in 1632. The viceroy placed
an epigraph in the town of Portici (in the Granatello area), inviting the local population to abandon
the Vesuvius area and recalling the catastrophic effects of the 1631 eruption. Many years later, for
this inscription, the expression "*the paradox of Granatello*" was coined by Nazzaro (2001). It refers
to the attitude of Vesuvius residents not to consider the risk (Nazzaro, 2001; Gugg, 2018). The
continuous activity of Vesuvius pushed many scholars and artists to visit the volcano (during the
famous Grand Tour epoch) and, at the urging of few intellectuals, the idea of founding a volcano
observatory gradually was born (Luongo, 1997). In particular, an important incentive to this idea
came from Sir William Hamilton (1730–1803), who arrived in Naples in 1764 as the British "Envoy
Extraordinary to the Kingdom of the Two Sicilies". Hamilton's amateur activity inspired the intuition
of active volcano surveillance and later, in 1841 (under the Bourbon Kingdom), the first
volcanological observatory in the world was founded, the Vesuvius Observatory (Cubellis et al.,
2015). It was a great moment for the Neapolitan School of Volcanology. In that period the interest of
this new institution was mainly devoted to the observation of the eruptive activity and to the
development of new instruments to monitor the volcano dynamic, such as the electromagnetic
seismograph designed by Luigi Palmieri (1855-1896) (Palmieri, 1880). Thus, the attention was
mainly posed on the volcanic hazard.
Later on, with the increase of population, the problem of volcanic risk became critical, because of the
exponential increase of the exposed value. The increase of population which experienced the
Neapolitan volcanic district was possibly sustainable, in respect to volcanic risk, up to the economic
boom of Italy, which followed the Second World War (Carlino, 2019). Immediately after this war
western civilization faced a long period of economic crisis. A global scale response to the crisis was
the activation of the Marshall Plan (the European Recovery Program, lasting from April 1948 to
December 1951), whose aim was the creation of stable economic conditions in order to guarantee the
survival of democratic institutions. The plan contributed to the renewal of the western European
chemical, engineering, and steel industries and to a rise in gross national products of between 15 and
25% (The Marshal Plain; https://www.history.com/topics/world-war-ii/marshall-plan-1). The
demographic increase in the province of Naples and the consequent expansion of urban areas since
the end of the Second World War have been largely influenced by the country's economic choices
following the Industrial Revolution, a process that began in the 19th century. For instance, the first
mechanical plants began in Pozzuoli in Campi Flegrei where, in 1885 a factory for the construction
of naval artillery was opened. The increase of population and postwar industrial activity mainly
involved the Vesuvius area, and in conjunction with the volcano's quiescent state following its most


recent eruption in 1944 (Carlino, 2019). The Campi Flegrei were also affected by a migratory flow
(albeit to a lesser extent) particularly in the districts of Fuorigrotta and Bagnoli (located inside the
caldera), where there was a strong phase of urban growth, especially following the expansion of the
Bagnoli industrial area in 1954 (Andriello et al., 1991). The social and environmental change within
the Campi Flegrei area had been drastic and often sudden but the area around Vesuvius was even
more badly affected. This latter came under attack from wild "cementification" not following any
town planning criteria, especially concerning the volcanic risk. In the westernmost sector of the
volcano, at the border with the eastern outskirts of Naples, oil refineries and various mechanical
industries were developed along the coastal strip, while between Portici and Torre Annunziata,
residential areas increased enormously (D'Aprile, 2014). Agricultural land in many areas was
converted into construction sites so that the landscape of farming and forestry use was transformed
into a typically urban, densely populated environment, clashing strongly with the background of
Vesuvius. Between 1950 and the 1990s, the entire Vesuvius area witnessed uncontrolled speculative
building with an exponential increase in residential areas, so as to make unrecognizable the
boundaries between the towns that, especially in the coastal sector, became merely an expanse of
housing and villas (Luongo, 1997; Carlino, 2019). In the whole metropolitan area belonging to Naples
an increase of 1,000,000 residents occurred between 1950 and 1980 (Censimento Popolazione Città
Metropolitana Napoli, 1861-2001). In this chaotic growth, the architectural beauties around Vesuvius
left over from the time of the Grand Tour, the historic villas were engulfed and new buildings covered
the lava flows arising from Vesuvius's most recent activity (Lancaster, 2008). This was a bad sign of
decline of local culture and of the corruption of political establishment (Berdini, 2010; Curci et al.,
230 2018).

With the onset of globalization and the expansion of international markets, the industrial activities in
the areas of Campi Flegrei proved bankrupt. This led to the definitive closure of Bagnoli's industrial
district in 1992 and to an attempt to reclaim the area, with numerous halts and changes in course, but
also taking place in the sector east of the city of Naples, closer to Vesuvius. Meanwhile, the
quiescence of Vesuvius, which has continued unbroken since 1944, gradually transformed the
volcano from a perceived condition of risk to that of a "passive" actor in the landscape. This step
resulted in inevitable demographic growth that did not tak the security implications into account while
the boom in the construction industry produced the extension of the cities around the volcano with
increasingly invasive settlements. Between 1950 and 1981, in the town of Portici alone, now one of
the most densely-populated places in the world, the population rose from just over 30,000 to about
84,000 (ISTAT Censimento popolazione e abitazioni). The extension of the cities around Vesuvius
took place centripetally, approaching more and more frequently the areas that have been repeatedly
affected by recent eruptions. If the quiescence of Vesuvius has caused a progressive decline in the


perception of volcanic risk, the territorial management policies until the end of the last century, have
continuously postponed to posterity the issue of the risks involved in spite of the continual efforts of
scientific community (Carlino et al., 2008). Only in relatively recent time, following the unrest which
affected the Campi Flegrei caldera in 1982-84, scientists, local authorities and the Civil Protection
faced the problem of excessive anthropic pressure in the Neapolitan volcanic area but an organic plan
for the decongestion one of the most areas of greatest volcanic risk is still lacking.

**3. The last experience of volcanic emergency in the Neapolitan district: Pozzuoli 1970-1984**
A fundamental moment in the history of volcano emergency in Campania is the episode of volcanic
unrest of Campi Flegrei caldera which affected the town of Pozzuoli in 1970-72 and 1982-84,
respectively. During those years the ground of the town experienced the maximum cumulative uplift
of about 3 meters, pushing the local authorities to evacuate the town, during both the episodes
(Barberi et al., 1984). By the beginning of the 1970s the phenomenon of *bradyseism* (a Greek origin
word which describes the up and down movement of the ground) was largely forgotten, since the last
time it had occurred more than 400 years before, when an uplift of about 20 m culminated in the
eruption of Monte Nuovo in 1538, the most recent volcanic event at Campi Flegrei (Di Vito et al.,
2016). In 1970 monitoring networks for volcano surveillance did not exist in the area, and the onset
of the uplift was initially observed by local fishermen. In fact, the inversion in the movement of the
ground, was signaled by fishermen, who suddenly managed to pass with their small boats beneath an
arch at the entrance of the small harbor of Pozzuoli while standing, while it had normally been
necessary to bend down (Carlino, 2019). The uplift, in the first phase, was almost aseismic, while the
Vesuvius Observatory, decided to undertake a new elevation survey, which was performed by the
engineers of the Genio Civile, to estimate the real amount of the ground uplift. The results showed
that the floor of the Serapeum of Pozzuoli (a ruin of an ancient Roman market) had risen by about
0,70 m since the last surveys, and that the area affected by this phenomenon included the entire town
(Luongo, 2013; Longo, 2019). The concern about the volcano uplift focused the attention on the
hazard related to a possible eruption. It was not a common opinion among scientists, thus, scientific
meetings took place to understand the way in which the phenomenon might evolve and the associated
volcanic risk. Experts like the volcanologists Alfred Rittman and Izumi Yokoyama participated in the
debate together with the researchers of Vesuvius Observatory. However, the physical model adopted
by the Japanese researchers associated the observed uplift with a high probability of an eruption. In
1972, the center of Pozzuoli was evacuated, although the unrest was characterized by a modest
seismic activity, while the maximum uplift was about 1.7 m and ended without eruption (Yokoyama,
1970). The evacuees were placed in the new Toiano district, whose construction was accelerated





during the final stages of bradyseismic episode. The 1970–72 bradyseism crisis, possibly was not
handled in a transparent way, and this experience was made more complex by the lack of sufficient
knowledge about the physics of the volcano phenomenon (Longo, 2019). This last fact possibly
determined the overcautious decision to evacuate the center of Pozzuoli. Nonetheless, it was during
that period that the Earth Science experienced new important studies and projects which also
strengthened the monitoring networks and the assessment of seismic and volcanic hazard in the
World.
Following the Campi Flegrei caldera unrest of 1970-72, the Italian peninsula was severely tested with
the devastating earthquakes of Friuli in 1976 (leaving about 1,000 people dead and more than 100,000
displaced) and the one in Campania-Basilicata in 1980 (with about 3,000 deaths and 280,000 dis-
placed) (Boschi and Bordieri, 1998). Subsequent to these events, a National Civil Protection service
was established in Italy. Thus, when a new bradyseismic crisis occurred in Pozzuoli in 1982, the
scientific community and the national and local authorities were better prepared to face the emergency
(Luongo, 2013). The Vesuvius Observatory had strengthened its surveillance network so that, over
the course of 1972-1981 it was possible to record a tendency to ground subsidence, and a new uplift
in 1982. In the summer of that year it became clear that a new episode of bradyseism was underway
(Cannatelli et al., 2020). This episode was most dramatic compared to the previous one. Continuous
and significant seismic activity was recorded since spring 1983. Pozzuoli was shaken by hundreds of
seismic events a day, while the population was frightened by the roars that accompanied the
earthquakes and the continued ground movements which wrought widespread damage on the city's
ancient buildings. A further increasing of seismic activity occurred between September and October
1983, reaching its peak on 4th October with a shallow magnitude 4.0 earthquake, causing panic
among the population, damaging several buildings in the historic center of Pozzuoli and being clearly
felt in Naples (Branno et al., 1984). The ground uplift in the Pozzuoli area reached a maximum rate
of the order of centimetres per day. The main concern about the situation was primarily related to the
building's damages caused by the shallow earthquakes (2-3 km in depth). Accordingly, the Vesuvius
Observatory and the National Group for Volcanology, responsible for surveillance, presented a
seismic hazard map of the Phlegraean area, showing that the level of risk in the historical center of
Pozzuoli had become very high, especially because of the high vulnerability of the buildings at risk
(Luongo, 2013). A further concern was related to the possibility of an eruption, for which the recorded
uplift and the seismic activity appeared as clear precursors, although the likelihood of an eruption
was considered low by the director of the Vesuvius Observatory. On 1st April 1984 a new dramatic
seismic crisis, with continuous swarms throughout the morning, hit the town of Pozzuoli. At this
stage, the problem of the evacuation was faced, also considering the possibility of an eruption
occurrence inside the caldera of Campi Flegrei. In collaboration with the Central Government, the





evacuation plan was drawn up and, following the meetings between monitoring staff and civil defense
authorities it was decided to evacuate about 25,000 people from the center of Pozzuoli. The evacuees
were relocated in the new settlement area of Monteruscello, which was built in few years, a few
kilometers north-west of the centre of Pozzuoli, considered a safer area than the coastal strip.
During the 1984 emergency, an effective communication system was established between the
monitors, the Civil Protection Service and the citizenry and the crisis was handled with maximum
transparency, especially in light of the 1970 experience (Luongo, 2013). In particular, the activation
of a monitoring info-center, close to Pozzuoli, was opened to ensure a correct management and
spreading of information about the ongoing events. Meanwhile, while the plan was actualized the
unrest seemed to decrease in intensity, and in December 1984 the uplifting and seismic activity
ceased, marking the end of the crisis (Barberi and Carapezza, 1996). Pozzuoli remained for few years
like a "ghost town", meanwhile local and central government were deciding about the future of the
city. Pozzuoli was later rebuilt without limiting the anthropic pressure that should have been
contained within thresholds that would make the volcanic risk acceptable. Today the municipality of
Pozzuoli has about 82,000 residents, and it represents a coveted residential site for Neapolitan people.

## 4. The debate about the volcanic risk in Neapolitan area
The subject of volcanic risk, and its mitigation, in Neapolitan area has very important implications
because this zone involves at least 1.500.000 people who are potentially exposed to a very large
eruption (Mastrolorenzo et al., 2006). Otherwise, giving the long history of volcanic risk in
Neapolitan area and the present very high risk of the area, two preliminary inquiries are required: i)
can we find a new paradigm or an alternative plan to reduce the high risk of the area? and ii) how is
it feasible in the Neapolitan area? We don't have a unique response to the questions but, to analyze
the issue, we have to go back again to the last Campi Flegrei caldera unrest occurred between 1982
and 1984, and culminated in the evacuation of the town of Pozzuoli (Barberi and Carapezza, 1996).
After this event a strong debate (among scientists, citizens and politicians) about the possible
solutions to reduce the volcanic risk in the densely inhabited Neapolitan area took place.
Between 1980 and 1990 the problem of volcanic risk in Neapolitan area was faced by the National
Group of Volcanology (GNV) (see De Vivo et al., 2010 and references therein), while the one of
territorial planning was discussed during several Italian workshops and few solutions were focused
on two main actions (Leone, 1987; Ulisse, 1984): i) the short-term one with the preparation of the
evacuation plans, ii) the long term one which provided the actions and methods aimed to reduce the
demographic pressure in the riskiest areas. As highlighted by Leone (1987), the latter is not a simply


action, because it doesn't represent a forced action, while it would be necessary to develop a new
organizational set-up of the whole Campania Region by planning a "new geography" of the services
industry and of the productive activities, allowing a spontaneous relocation of the residents from the
risk areas.
After the last Campi Flegrei caldera unrest, ended in 1984, the volcano became rests again (up to
2005), but not the debate about volcanic risk. Later, to respond to the solicitations and concerns
coming from scientific and institutional world, and following the foundation of the Italian Civil
Protection, the attention was mainly posed on the Vesuvius, which is the most inhabited volcano of
the district. The volcanic risk in this area was evaluated by Scandone et al., (1993), in terms of human
losses, and according to the equation: *Risk = Exposed Value × Vulnerability × Hazard* (Blong, 1996).
The authors evaluated hazard based on the entire history of the volcano and identified the events
likely to cause loss of human lives as those with VEI>~3. Later on, the first evacuation plan for the
Vesuvius area was released by the Civil Protection in 1995.
After its foundation in 1999, the INGV (Istituto Nazionale di Geofisica e Vulcanolgia) became the
reference scientific institution for the Civil Protection, to provide the assessment of volcanic hazard
and its continuous updating for Neapolitan volcanoes. As regard the Vesuvius, the extension of the
most hazardous zone (i.e. the Red Zone) involves about 600,000 inhabitants which must be evacuated
in case of eruption (Protezione Civile: Update of the National Emergency Plan for Vesuvius). The
extension of the Red Zone was obtained considering a medium energy scenario for the next eruption
(a sub-plinian eruption) like that occurred in 1631.  The emergency plan for Vesuvius foresees, that
a part of the population spontaneously moves away from the Red Zone during the pre-alarm phase
(Fig. 1). Depending on the state of the volcano, the actions to be taken are defined within the
emergency plan by the different levels of alert, in which the scientific and monitoring activities are
decided upon depending on the assessment of the hazard. The lowest level (a "green" alert level)
corresponds to the quiescence of the volcano, during which there are no significant changes in the
parameters being monitored. If these changes are detected however, the protocol provides for a
transition to a level of attention ("yellow"), during which there is an intensification of monitoring
activities and a more frequent assessment of the condition of the volcano by the Civil Protection
agency and the Italian Commissione Grandi Rischi (Major Risks Commission). The levels above this
are those of pre-alarm ("orange") and alarm ("red"), which, for the latter, involves the evacuation of
the population from the Red Zone. The Vesuvius evacuation plan has been updated and modified
during the time. At the present, at least three days (compared to the previous three weeks) would be
required to allow the effective evacuation of 600,000 inhabitants. This should correspond to the actual
possibility of forecasting the eruption with this level of forewarning. The last choice was also based



on the forecasting experiences of the 1980 Mt. Saint Helens (USA) and 1991 Pinatubo (Philippine)
eruptions (Swanson et al, 1983; Pinatubo Volcano Observatory Team, 1991). The plan posed, among
the scientific community, a number of concerns and criticisms about the actual possibility of
forecasting the next eruption in advance and evacuate at least 600,000 people at risk. In the framework
of this debate, an alternative plan to mitigate the volcanic risk of Vesuvius area was proposed by
Flavio Dobran (*Vesuvius 2000* plan, Dobran 2006, 2007). Although the first work of Flavio Dobran
was published in 2006, the dissemination of his plan took place few years earlier, with an intense
information campaign around the Vesuvius area. More than an emergency or evacuation plan,
*Vesuvius 2000* was a proposal of a new paradigm of development to reduce the risk of the area. The
main intention of this proposal was "…*to produce guidelines for transforming high-risk areas around*
*Vesuvius into safe and prosperous communities. This would be accomplished through*
*interdisciplinary projects involving engineers, environmentalists, urban planners, economists,*
*educators, geologists, sociologists, historians, and the public*" (Dobran, 2007). Among the general
aim of *Vesuvius 2000* plane, the decreasing of the resident population density in the most-risky areas
was proposed, as well as improving of the resistance of the buildings, the quality of infrastructure and
the resilience of urban centers. Furthermore, Dobran (2006, 2007) showed that, giving the strong
historical and social connection between "Vesuvius people" and their land, the lightening of urban
pressure in most of the risky zones represented a very long-term aim, which needs a complete social,
cultural, urbanistic and economic reconsideration of the Vesuvius area and surroundings. This long-
term action will minimize the economic and social costs due to evacuation of people from the red
zone in case of eruption. The great challenge of the ambitious *Vesuvius 2000* plan, was therefore that
people living around the volcano acquired the awareness of the environment in which they live and
participated in the solution of this difficult situation (Dobran, 2006).
Behind the solution proposed by Dobran (2006, 2007), a wide literature about the methods and the
actions devoted to reduction and management of volcanic risk, and also of natural risks in general,
was proposed by different authors, and in which most detailed descriptions of the limits of each
solution and the cases history are reported (Peterson et al, 1993; Newhall and Punongbayan, 1996;
Chester et al., 2000; Small and Naumann, 2001; Petrazzuoli and Zuccaro, 2004; Wisner, 2003;
Petrosino et al., 2004; Spence et al., 2007; Hansjürgens et al., 2008; Barcklay et al., 2008; 2015;
Jenkins and Haynes, 2011; Usamah and Haynes, 2012; Hicks et al., 2014; Hossain et al., 2017;
Fearnley et al., 2017; Papale, 2017). Furthermore, some of the above researches also demonstrate that
a volcanic resettlement program must be directed by meaningful consultation with the impacted
community, as also suggested by Dobran (2006), who also shares the decision making.



What happened in the period following the first releasing of the Vesuvius emergency plan and of the
alternative paradigm *Vesuvius2000* proposed by Flavio Dobran? The latter was not welcomed to the
political establishment and remained a mere proposal. On the other hand, the former (the institutional
one) only partially guaranteed the restraint or decreasing of anthropic pressure around the volcano.
To deal with this problem, a new plan called *Vesuvìa* (https://www.viveretraivulcani.it/il-progetto-
vesuvia/), was approved in 2003 by the Campania Region (Legge regionale n. 21/2003, "Legge del
Vesuvio", http://www.sito.regione.campania.it/leggi_regionali2003/lr21_2003.htm). The intent of
this project was to lighten the demographic pressure around the Vesuvius volcano. This intent would
be promoted by offering economic incentives (up to 30 thousand euros) to the population (living in
the red zone) willing to relocate themselves outside the dangerous areas. The project expected to
reduce the number of people living in the red zone over a period of about 20 years by removing at
least 100,000 people from this zone (Gugg, 2018). A further aim of *Vesuvìa* was also the reconversion
of available buildings into tourist reception facilities, in order to create an opportunity of valorization
of the great cultural and natural heritage of the Vesuvius volcano. (http://www.cngeologi.it/wp-
content/uploads/2017/08/Casa-Italia_Rapporto-sicurezza-rischi    naturali-patrimonio-abitativo.pdf).
After three years from the launch of the project there was a reduction of residents in the red zone of
only 0.1%, moving the promoters of the project to leave the endeavor. Actually, it was a flop. The
reasons of the failure were described by Gugg (2018). Among the reasons reported by the author, the
lack of involvement of the majors and the local communities in the development of the project was
probably the most critical for its flop. Additionally, as also described by the *Vesuvius 2000* plan
(Dobran 2006, 2007), a relocation of people from the red zone outside the Vesuvius volcano is very
unlikely lacking a long-term economic and social policies which stimulate Vesuvius people to move
in safer zones. It is clear that in a complex social, cultural and urban context like that of Naples and
surroundings, the choice to reduce the volcanic risk by relocating a part of people living in the red
zones (Campi Flegrei and Vesuvius) outside the most-risky areas and by increasing the volcanic
perception is a very grueling challenge (Carlino, 2019). Furthermore, the policies to improve the
vulnerability of edifices against disasters (and reduce the risk) have been rarely adopted in Italy, as
demonstrated for instance by heavy damages suffered by many cities after moderate earthquakes
occurred in recent times (Valensise et al., 2017). The main issues, in this case, are related to the actual
perception of risk in general (as well as of volcanic risk in particular), but mainly to the morals and
personal profit of politicians in doing specific actions to reduce the risk and to other social and
political problems of the Neapolitan area (Luongo, 1997; Carlino et al., 2008; Donovan and
Oppenheimer, 2015; Donovan, 2019). For instance, political timescales generally limit the amount of
capital that is invested in the volcanic risk reduction. Basically; as reported by Donovan (2019), "*if a*
*politician is only in power for 4 years*" (and this time is the best case in Italy!) "*the probability of an*
*eruption at a particular volcano within that timeframe is usually very low, and so, the personal-*



*political cost-benefit analysis indicates that there are more socially acceptable policies to invest in*".
This is possibly one of the main reasons why a long-term plan for the risk reduction such as the one
of *Vesuvius2000* was refused by political establishment. The example reported by Donovan (2019)
appears particularly true for the Neapolitan area, where the volcanic risk increased exponentially
during the last 50 years and no policies actions have contained this trend. This aspect was also debated
by De Vivo et al., (2010) who stated that while the Italian Civil Protection tries to convince people
to dislocate from the risk zone, at the same time it does not take a stand against the illegal buildings
in the red zone. Otherwise, from the institutional point of view, the latter problem does not involve
the Civil Protection, because the management control of illegal buildings and their compliance in
respect to the seismic risk primarily involves the municipalities (*Decreto Legislativo 18 agosto 2000,*
*n. 267; Testo unico delle disposizioni legislative e regolamentari in materia edilizia, d.P.R. n.*
*380/2001*). In this regard, the seismic risk associated to the volcano-tectonics earthquakes is not
neglectable as well, at least for Campi Flegrei and Ischia. An representative case is the Island of
Ischia. In 1883 the island was hit by a moderate and shallow earthquake (with magnitude around 4.5,
Cubellis and Luongo, 1998) which devastated its northern sector (Casamicciola town) and caused
more than 2300 victims (Carlino et al., 2010b). This event was followed by an almost seismic silence,
up to 2017. At least during the last 25 years the scientific community stimulated the island local
authorities and the municipality of Casamicciola to take actions in favor of the mitigation of seismic
risk in the island (Cubellis and Luongo, 1998; Luongo et al., 2012). But this message went unnoticed,
up to the 21 August 2017, when a $M_L 4.0$ earthquakes occurred in Casamicciola town and caused 2
victims, tens of injuries and heavy damage in the upper part of the municipality (De Novellis et al.,
2018). Form the above considerations, it appears that conciliating the emergency plans, the drawing
of the red zones of volcanoes, and the regulations for the seismic risk, with the actual economic and
land-use planning policies in Neapolitan area is a hard purpose to attain.
Recently, in August 2016, the emergency planning for the volcanic risk of the Campi Flegrei was
updated (Protezione Civile: Update of the National Emergency Plan for Campi Flegrei), and the area
of the new Red Zone to be evacuated as a precautionary measure in case of eruption, was defined,
together with the Yellow Zone, that is potentially exposed to a high concentration of falling ash (Fig.
1). As for Vesuvius, the Red Zone and the Yellow Zone were defined by the Civil Protection, in
agreement with the Campania Region, and based on the indications provided by the scientific
community. As a whole, and considering that the emergency plan for the island of Ischia (Gulf of
Naples) is still lacking, about 1,000,000 of people could be directly affected by a moderate to large
eruption (VEI 3-4) in the red zones of Campi Flegrei and Vesuvius, respectively. The high number
of people exposed to the risk, and the uncertainty in eruptions forecasting (Sparks, 2003), pushed
some authors to criticize the evacuation plans and the policies of risk reduction in Neapolitan district


(Rolandi, 2010; De Natale et al., 2020). In particular and recently, De Natale et al., (2020) have
questioned about how the very high volcanic risk in the Neapolitan area can be effectively mitigated.
The authors focused the attention on two problems related to the evacuation: i) the extremely high
number of people to evacuate in case of an impending eruption; ii) the lack of plans today to reallocate
such a high number of evacuated people (600,000 and 700,000 for Campi Flegrei Caldera and
Vesuvius, respectively). The analysis of De Natale et al., (2020) is not new, since their main
conclusions, as well as and the weak points they highlighted in respect to the present emergency
plans, were already stated by other authors, and in particular by Dobran (2006, 2007, *Vesuvius 2000*
plan). It is important to highlight that some works criticizing the evacuation plans (Dobran 2006; De
Natale et al., 2020), do not exclude their effectiveness if a number of actions to mitigate the risk is
carried on. Unfortunately, what we have seen during the last 40 years of volcanic risk management
in Neapolitan area, is a predominance of the emergency policies in the respect to that of prevention.
The result is that the present volcanic risk, giving the current high values of society, appears non-
acceptable.

**5.  The role of volcanologists**
In the framework of the discussed topics a fundamental issue is the role that volcanologists must have
in managing volcanic risk and volcanic crises. It was, in many cases, misinterpreted by people living
in Neapolitan area. The role and responsibilities of volcanologists in volcanic hazard evaluation, risk
mitigation, and crisis response have been treated by the International Association for Volcanology
and Chemistry of the Earth's Interior (IAVCEI). Their main responsibility is to improve the scientific
knowledge of volcanoes to better understand how they work and provide most robust eruption
forecasts, and to educate the local and global community (mainly exposed to eruptions) to the
volcanic risk, making people more perceptive against the risk itself. The latter is fundamental to get
a good response from people to an evacuation (IAVCEI, 2016). Anyway, the main task of
volcanologists remains to provide a forecast as more robust as possible of an eruption. It is well
known how problematic it is to obtain a clear picture about the progression of volcano processes
during unrests and to understand which is the actual state of the volcano (critical state or not). In
general (but not always), as the eruption is approaching the number and amplitude (or energy) of
geophysical and geochemical signals increases and the uncertainty in the forecast should decrease
(Decker, 1986; Kilburn, 2003; Sparks, 2003; Robertson et al., 2016; Sparks and Cashman, 2017;
Carlino, 2019) (Fig.4). An unsolved question is whether, and in which moment, the volcano
approaches the critical state during an unrest, that is the moment in which the physical processes
occurring within the volcano are irreversible, and the volcano will erupt (Fig. 4). This is the most
critical issue, because the promulgation of a false alarm or a missed alarm, will adversely affect
600,000 to 1,500,000 of people leaving in Neapolitan area (De Natale et al., 2020). During the last
20 years, the monitoring networks for the surveillance of Vesuvius, Campi Flegrei and Ischia
volcanoes have been greatly improved, reaching one of the best standard worldwide
(www.ov.ingv.it). This effort should correspond to a reduction of the uncertainty in forecasting the
next eruption, although it depends on the capacity of volcanologists to correctly decipher the volcano
signals. Beyond the efforts of scientists to improve their understanding of volcanic processes and
providing more robust forecasts, it is fundamental to communicate the systemic uncertainty of the
forecast to the public. This can be done in an effective fashion only if a proficient direct
communication network between volcanologists and the media is provided (Haynes et al, 2008). This
is also a very important topic, particularly when the communication of an ongoing volcanic crisis
involves large metropolitans' areas like Naples and surroundings. The example of what occurred
during the 1982-84 unrest is emblematic in this view. During that crisis a unique channel of
communication was established between the direction of Vesuvius Observatory and the press, while
the observatory was continuously in communication with the Minister for the Coordination of the
Civil Protection (Luongo, 2013). The activation of the information center for the citizens of Pozzuoli
and the straight link between the latter and the direction of the Vesuvius Observatory, generated more
confidence among people. How would it have gone if the same crisis had happened today? The unrest
and the evacuation at Pozzuoli occurred in a period without internet and social media (like Facebook,
Twitter and WhatsApp) which, nowadays, represent the main and quicker dissemination channels of
news and information. The social media are a disruptor of traditional communication, opening up
new opportunities for scientists to communicate (Dong et al., 2020) but, on the other side, giving the
right to evaluate or criticize scientific decisions to everyone. This could lead to misinterpretations or
distortions of scientific broadcasts and information and, consequently, to false alarms or unjustified
panic among the population, in case of a volcanic crisis. This circumstance, albeit not related to a
volcanic crisis, occurred in recent time, before the starting of the Campi Flegrei Deep Drilling Project,
at Campi Flegrei, a project aimed to scientifically investigate the caldera (Carlino, 2019). The project
worried many local residents about the possible disturbance which the scientific drilling would cause
on the volcanic system. Just before the onset of the drilling, the declarations that continued to spread
on social networks and newspapers became increasingly catastrophic (sometimes at the limit of the
paradoxical) such as to seriously worry the municipal administration of Naples, which had issued
clearance for drilling. The climax was reached on October 2010, when the national newspaper "Il
Mattino" led with the front-page title: "If you touch the volcano Naples will explode" (Carlino, 2019).
The project was temporarily suspended by the administration of Naples to further clarify its aim and
associated risk. This fact highlights that the position of volcanologist in communicating the hazard
and the risk in densely inhabited areas like Naples, is very tricky because the communication occurs


within a complex social system where many people exposed to the risk are involved. Furthermore, a
number of studies demonstrates that Neapolitan people have a low perception of risk and a low level
of risk education (Carlino et al., 2010b; Ricci et al. 2013).
As a whole, beyond the effort that scientists are sinking to improve the robustness of volcanic
eruptions forecast, a further effort is necessary to promulgate the culture of volcanic risk and promote
open debates with the local population and authorities. In other words, volcanologists should be more
present on the territory (not only during an ongoing volcanic unrest) and they should be an open book,
not an acquired skill (Goodstain, 2010; Fearnley et al., 2017). This approach is fundamental to
improve the confidence of people in a scientific institution such that of INGV.

## 564    6. Conclusions

The past experiences concerning the management of volcanic risk in Neapolitan area reveal how
complex is to devise a collaboration around the active volcanoes of Vesuvius, Campi Flegrei caldera
and Ischia Island to reduce the risk in such densely inhabited areas. The history of volcanic risk in
this area demonstrates the leaning not to consider, or to underestimate, the risk (which otherwise is
an attitude of human being). Nonetheless, we cannot constrain the problem of the high volcanic risk
of Neapolitan area to this latter consideration only. The present development of the urbanized areas
around the volcanoes of Naples is the result of a very long history and stratification of different
cultures and population which settled the Neapolitan area and its surroundings as a nice and useful
place to live, since the Bronze Age. This history left a huge cultural heritage but also a difficult socio-
economy condition, especially around Vesuvius. Thus, as also highlighted by Galliard (2008), in
many cases the historical and cultural heritage and political-economy remain of much greater
importance and may overcome the choice of people in the face of volcanic hazards. This fact
emphasizes the importance of understanding the larger and daily contexts of Neapolitan area in
proposing the policies to reduce the volcanic risk. It appears evident, for instance, that the choice of
people not to relocate themselves outside the red zone of Vesuvius, and to remain in their native
towns, despite the perceived threats, has little to do with volcanic activity. This point, already
discussed by Galliard (2008), suggests that, in such a complex social context (i.e. the Neapolitan
area), the policies for volcanic risk mitigation need to go far beyond the only prevention of relatively
rare events. A different and more general approach is thus required and it should be aimed to a rational
access and use of resources in order to adapt the social and economic development of the area to its
natural vocation. This is a long-term objective which conflicts with the short-lived (and not forward-
thinking) policies adopted by the Campania Region and the Central Government. Consequently, the



proposals to re-convert the riskiest areas of Neapolitan volcanoes into lower risk zones using a
different (and long-term) paradigm of development (e.g. Dobran, 2006, 2007), are struggling taking
off. At the same time, the proposed economic-incentives (*Vesuvìa* project) to relocate people from
the red zone (at Vesuvius) towards more safety areas was a failure as well. Accordingly, these failures
first have to do with a wrong territorial policy, and secondly with the volcanology.
Furthermore, at least during the last 25 years, the policies for the reduction of volcanic risk in
Neapolitan area have been disconnected from their natural, social and politico-economic context.
This is possible the result of a not holistic approach to the problem of volcanic risk reduction which,
in particular in this area, is unavoidable and, on the contrary, requires an openly discuss method
between academics of all disciplines, policymakers, and stakeholders (Dovovan, 2019). Finally, after
about 40 years of debates around the volcanic risk in Neapolitan area, an analysis of the reasons why
the strategies aimed to reduce the risk in this area were systematically failed is required. This step is
necessary to propose more reliable solutions for the risk reduction in a very complex territory like
that of Neapolitan volcanoes. A further effort is also required by Neapolitan scientists to connect the
territorial governance structures and local (at risk) communities with the scientific network. In this
framework, a further attention of scientists must be addressed to avoid to politicize the volcanology
in advising authorities (Donovan, 2019).
***Data availability***: No datasets were used in this article.
***Competing interests of interest***: The author declares that he has no conflict
**Figure captions**
Fig.1. The Neapolitan volcanic district with the three active volcanoes, the Vesuvius, the Campi
Flegrei caldera and the Island of Ischia. The limits of the red zones of the evacuation plans for
Vesuvius and Campi Flegrei caldera are reported, respectively (from www.protezionecivile.gov). A
whole of more than 1,000,000 of people are living in both the red zones. A plan for the island of
Ischia is currently in progress (base map is from Google Earth).
Fig. 2. A summary of the volcanic activity history at Vesuvius, Campi Flegrei and Ischia Island (from
Piochi et al., 2005).
Fig. 3. The city of Naples with the location of the eruptive vents associated with different eruptive
periods. The dotted line represents the eastern boundary of the caldera of Campi Flegrei (modified
after Scarpati et al., 2013 and Carlino, 2019; base map is from Google Earth).




Fig. 4. A qualitative sketch describing the possible state of a volcano approaching an eruption its
forecast reliability. For a quiescent volcano the reawakening is generally associated with the onset of
seismic signals that mark the variation of stress field within the volcano, the circulation of pressurized
fluids and, eventually, the magma migration at shallow level. This dynamic is accompanied by others
precursors (ground deformations and variation of fluids emission) which make the forecast more
reliable as the eruption is approached. The point at which the volcano overcomes the critical state, is
the moment (t?) in which the physical processes occurring within the volcano are irreversible, that is
to say the volcano will erupt. Volcanologists cannot predict the time (t?) because the processes are
chaotic and the forecast has a probabilistic nature (after, Carlino, 2019).
**Figures**

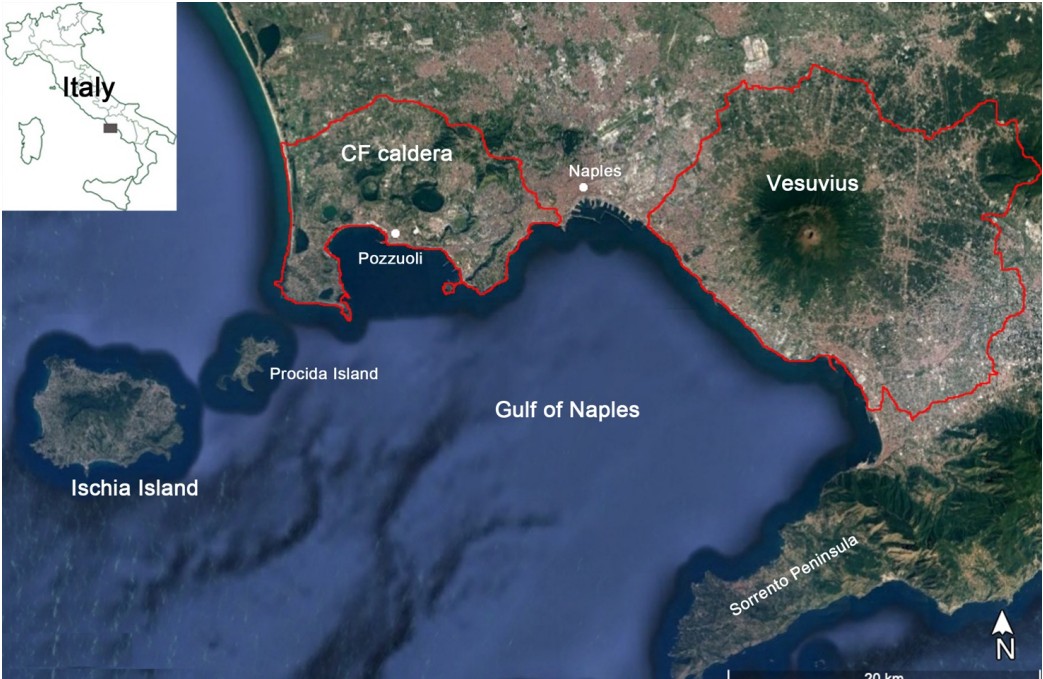


**Fig.1**

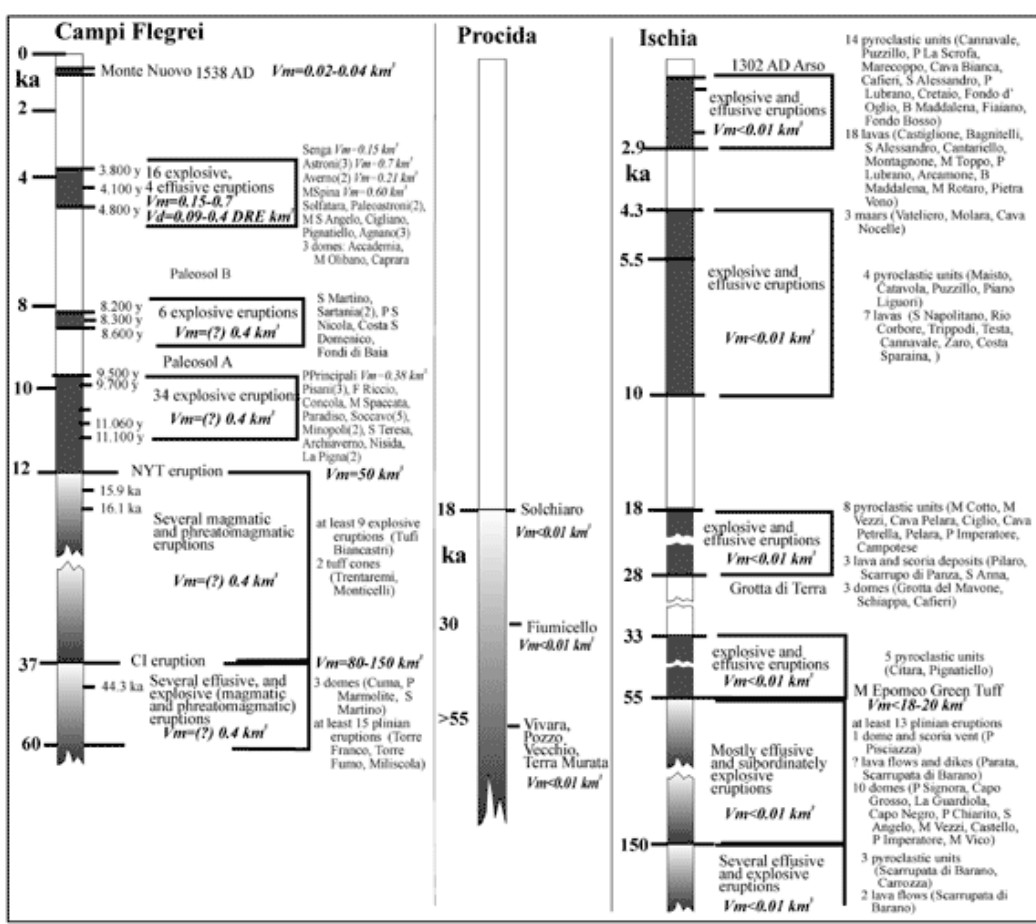

**Fig.2**




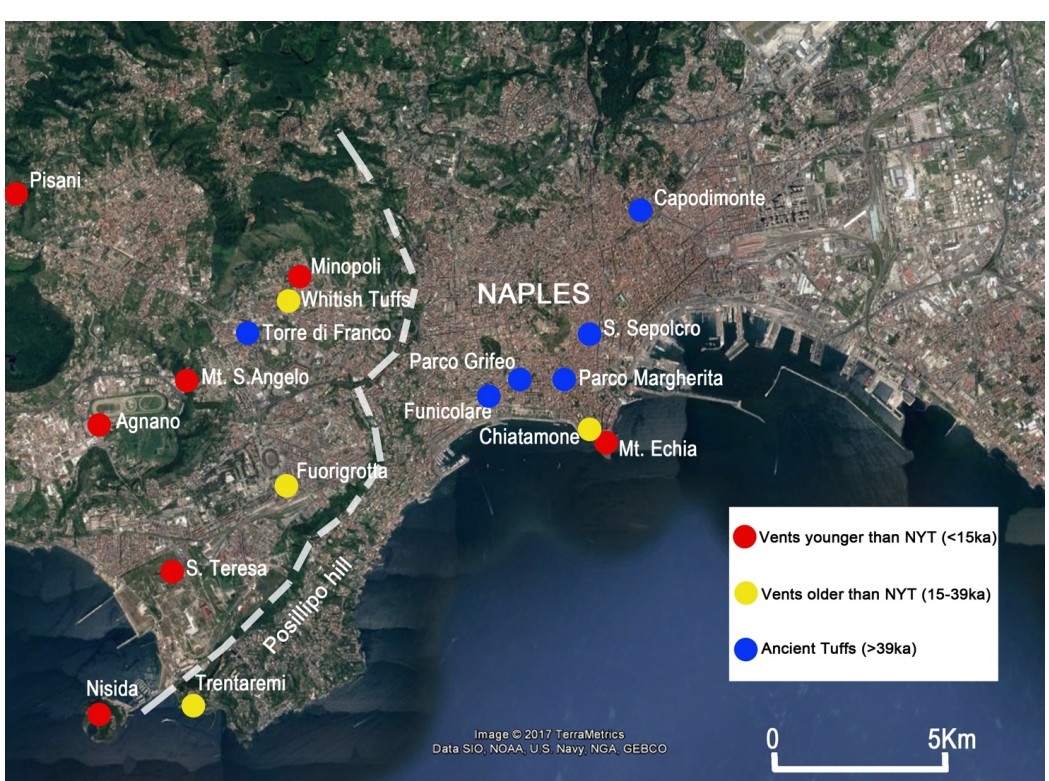

**Fig. 3**

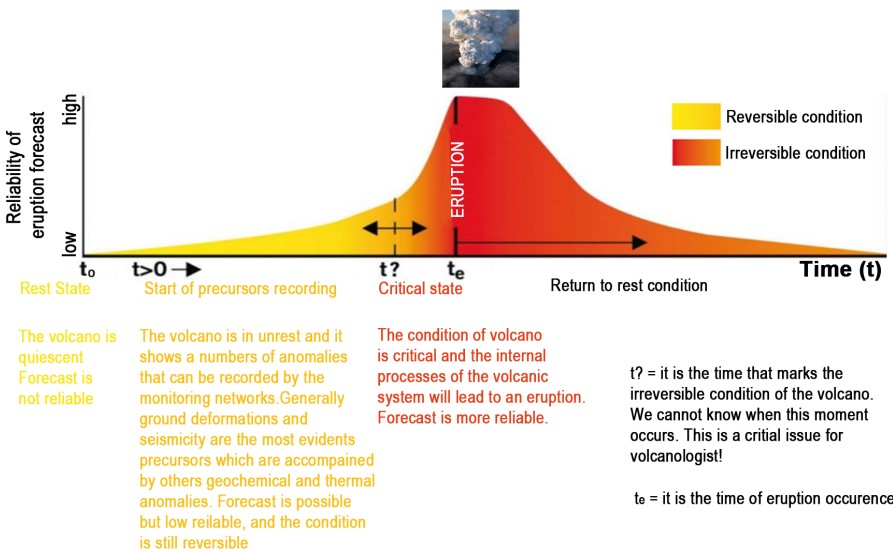

**Fig.4**





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
