# Peer review of "Review article: Brief history of volcanic risk in Neapolitan area (Campania, Southern Italy): a critical review Stefano Carlino"

_Natural Hazards and Earth System Sciences, 2020_

## Author Response (AR1)

Reviewer1

The text is not suitable for publication as it stands because the standard of written English is not up to publishable standard. The whole manuiscript needs to be re-written. I started to correct the manuscript using 'track change' but it was taking me a hour per page and this is clearly not the role of the reviewer.

The academic content is fine and the paper should be published, but not in its present form.

**Dear Reviewer**

**thanks for your comments. The English of the paper was revised by professional editing service and also a few contents of the paper have been rewritten according to the comments of the Reviewer2. Attached you can find the tracking file with all changes. The parts that are rewritten, following the comments of the Reviewer2, have been highlighted with yellow color in the tracking file. I hope that the new version of the paper will be suitable for the publication to NHESS.**
**Kind regards**
**Stefano Carlino**

**Reviewer2**
I agree with R1 – the paper is written in much better English than my Italian, but it is hard to read and in places the style is colloquial/journalistic rather than academic. I have done a quick read-through with some other points to note but will review it in more detail after a rewrite. I also agree with R1 that there is much excellent material here and strongly encourage the author to persevere with it!

**Dear Reviewer2**

**thanks for your suggestions. Accordingly, the English was revised by professional editing service and further changes have been done following the points you highlighted. Below, you can find the point-to-point responses, while attached you can find the tracking file with changes. The parts that are rewritten, according to your comments, have been highlighted with yellow color in the tracking file.**

Other points:

How do we know that people were so attracted to the area by the volcanoes? There are lots of highly populated non-volcanic areas in the region. There are comments throughout like this – that make value judgements with limited evidence – e.g. page 5 has quite a simplistic reading of culture as sequential.

**Reply: I explained this point using a different preliminary consideration. However, people did not settle this area (the Neapolitan district) simply because they were attracted by volcanoes. Actually, volcanoes created the natural environmental conditions, such as the high fertility of the soils for the agriculture and the presence of thermal waters, lakes and natural inlets, which were favorable condition for the development of a local "economy" and also for leisure (the latter aspect was particularly appreciated by Romans during their Empire). On the other side, up to the Middle Age, volcano disasters were considered a punishment for human beings and thus the concept of hazard and risk was not related to the Earth natural cycles, on the contrary it was associated to the myth. This belief generated and underestimation of the actual volcanic**

risk. Finally, since large and destructive eruptions are rare events, during the development of the stable settlements of the area people considered that the benefit of living around active volcanoes was greater than the risk associated to an eruption.

Ll47-48 Not really an academic comment!

**Reply: yes, I changed this comment**

Ll61 onwards – not always necessary (or indeed possible) to quantify vulnerability – vulnerability needs to be dealt with in different ways. Quantification can help but is not the only approach. The primary drivers of vulnerability may be socio-economic, cultural and political, and so policy changes and reducing social inequality are more important than measuring vulnerability itself.

**Reply: I agree with this comment. I changed this part of the paper according to your suggestion.**

On the C17th, there is a useful book by Sean Cocco

Cocco, S., 2012. Watching Vesuvius: a history of science and culture in early modern Italy. University of Chicago Press.

**Reply: I know very well this book. It was my mistake not to cite it. I added this book in the reference list and in the text.**

L281: whether or not it was "overcautious" to evacuate Pozzuoli depends also on the uncertainty – it is not just about what happened, but what could have happened – if the uncertainty is high, the evacuation may be justified anyway.

**Reply: This is true. Otherwise, I wanted to underline that, during the first unrest at Campi Flegrei, in 1970-72, the general knowledge of volcano dynamic was modest and the monitoring of Campi Flegrei caldera was virtually absent. This was possibly the main reason that led to the hasty choice, to begin the evacuation of the population of Pozzuoli. This history is interesting, since there was a suspicion that an attempt at building speculation was at the heart of this choice. In fact, more than an evacuation, there was a forced eviction of the inhabitants of Rione Terra (the historical center of Pozzuoli), who were temporarily placed in hotels and hospitals, awaiting their definitive transfer to the new residential district, the Rione Toiano district, which was to have been built a few kilometers north of Pozzuoli. Amidst the bewilderment of a population besieged by the police and the chaos caused by the closure of many of the access routes to the city, the suspicion of a building speculation manoeuvre became a conviction for many residents. Anyway, I avoided to use the term "overcautious" and explained better this point in the new version of the paper.**

Some of the information in this section (historical activity of CF) could be displayed in a timeline, which would be helpful for readers unfamiliar with the events. The existing figures are very good – would just be useful to have a timeline of the more recent crises/unrest too.

**Reply: according to your suggestion a figure with the timeline of the main volcanic events have been added. It replaces the figure 2 of the previous version of the paper**

**Kind regards**

**Stefano Carlino**

---

## Author Response (AR2)

Dear Amy Donovan

Many thanks for your further comments to the paper. I tried to accommodate all your suggestions and comments that you can see in the new submitted version.

1.  I have split the section 2 into three different sections. Section 2 is: "The first human settlements of Neapolitan volcanoes" and it shows the ancient history of human settlements; Section 3 is: "Towards a modern view of volcanoes" it shows how the change of scientific paradigm (introduced by Galileo Galilei) influenced the development of volcanology, particularly in Neapolitan school; Section 4 is: "Volcanic risk increase" and it shows how the risk was increased particularly starting from the post-Second World War era. I have also split the new section 7 (old section 5) into one section and sub-section related to the role of volcanologists and to volcanologist and media, respectively.

2.  I did not add a new section, while I preferred to insert in the conclusion your considerations about disaster-development trajectories and mistakes that are being repeated all over the world in hazard-prone areas. I hope you will agree with this choice (see lines 609-614 of the new version).

3.  I added a sentence about the considerations of Dave Chester in the new version (please see line 176-179). I also added the two references you mentioned in the comment.

4.  I added a new sentence (and the reference) about the work of Annie Winson. You can find it in the new version of the paper (lines 525-529).

5.  A population density map is inserted in the figure 1. I tried many times to download a population density map from the GHS dataset, but in opening the file an error always appears. I used a population density map of Regione Campania. I hope it works as well.

6.  I added both the references of Harris in the section 7 and 7.1. I also added a sentence of Harris 2015 in the lines 549-551.

7.  L25, 433, 469 and 497 have been modified as you requested.

Best regards